# Rho GTPases in the Physiology and Pathophysiology of Peripheral Sensory Neurons

**DOI:** 10.3390/cells8060591

**Published:** 2019-06-15

**Authors:** Theodora Kalpachidou, Lisa Spiecker, Michaela Kress, Serena Quarta

**Affiliations:** Division of Physiology, Department of Physiology and Biomedical Physics, Medical University of Innsbruck, 6020 Innsbruck, Austria; theodora.kalpachidou@i-med.ac.at (T.K.); lisa.spiecker@i-med.ac.at (L.S.)

**Keywords:** Rho GTPases, actin cytoskeleton, sensory neurons, neurite outgrowth, neuroregeneration, development, inflammation, pain

## Abstract

Numerous experimental studies demonstrate that the Ras homolog family of guanosine triphosphate hydrolases (Rho GTPases) Ras homolog family member A (RhoA), Ras-related C3 botulinum toxin substrate 1 (Rac1) and cell division cycle 42 (Cdc42) are important regulators in somatosensory neurons, where they elicit changes in the cellular cytoskeleton and are involved in diverse biological processes during development, differentiation, survival and regeneration. This review summarizes the status of research regarding the expression and the role of the Rho GTPases in peripheral sensory neurons and how these small proteins are involved in development and outgrowth of sensory neurons, as well as in neuronal regeneration after injury, inflammation and pain perception. In sensory neurons, Rho GTPases are activated by various extracellular signals through membrane receptors and elicit their action through a wide range of downstream effectors, such as Rho-associated protein kinase (ROCK), phosphoinositide 3-kinase (PI3K) or mixed-lineage kinase (MLK). While RhoA is implicated in the assembly of stress fibres and focal adhesions and inhibits neuronal outgrowth through growth cone collapse, Rac1 and Cdc42 promote neuronal development, differentiation and neuroregeneration. The functions of Rho GTPases are critically important in the peripheral somatosensory system; however, their signalling interconnections and partially antagonistic actions are not yet fully understood.

## 1. Introduction

The members of the Ras homolog family of guanosine triphosphate hydrolases (Rho GTPases) are small GTP binding and hydrolysing proteins of approximately 21 kDa, which together with ADP ribosylation factors (Arfs), Ras-related proteins in brain (Rab), Ras-related nuclear protein (Ran) and Ras belong to the Ras superfamily of small GTPases [1]. In humans the Rho GTPase family has about 20 members that can be subdivided in to classic (typical) and atypical GTPases (Table 1) [2,3,4] and the most well studied today are Ras homolog family member A (RhoA), Ras-related C3 botulinum toxin substrate 1 (Rac1) and cell division cycle 42 [Cdc42 (Figure 1)].

The classic Rho GTPases hydrolyse GTP to GDP and thus cycle between the GTP bound active and the GDP bound inactive state [1]. The GTP/GDP cycling mechanism is finely tuned by Rho-specific guanine nucleotide exchange factors (GEFs), which promote the active state and GTPase activating proteins (GAPs), which favour the inactive state [5]. Additionally, the membrane localization of classic Rho GTPases affects their activity, and this is controlled by guanine nucleotide dissociation inhibitors (GDIs) [6]. In contrast, the atypical Rho GTPases are constantly bound to GTP, they do not hydrolyse GTP and there are no data supporting their regulation by GEFs or GAPs [3,7].

The cellular distribution of Rho GTPases indirectly regulates their function by restricting them to certain subcellular compartments. The intracellular localization of Rho GTPases is regulated by post-translational modifications (PTMs), such as isoprenylation [20,21], which provides a membrane anchor or palmitoylation [22,23]. Moreover, the presence of a functional nuclear localization signal (NLS) sequence permits the correct nuclear entry and accumulation of these proteins [24]. Rho GTPase PTMs, such as phosphorylation, ubiquitylation and sumoylation, not only determine their localization but may also directly affect their function [25]. Various kinases, such as cAMP-dependent protein kinase A (PKA), cGMP-dependent protein kinase G (PKG), Src kinases and Akt, directly target and phosphorylate GTPases [25]. Phosphorylation changes the GTPase binding affinity to guanine nucleotides, promotes dissociation from the membrane and even induces degradation [25]. Ubiquitylation induces degradation of Rho GTPases [26], whereas Rac1 sumoylation increases Rho GTPase activity [27]. Additionally, different cytotoxins can either deactivate Rho GTPases via ADP-ribosylation, glucosylation, glucosaminylation and AMPylation or activate them via transglutamination [28]. Rho GTPase expression can be also regulated post-transcriptionally by microRNAs (miRNAs), such as miR-124 [29].

Rho GTPases can be activated by various extracellular signals acting on their respective receptors, such as G-protein coupled receptors (GPCRs) [30], receptors of the tyrosine kinase (RTKs) family [31], ionotropic receptors [32], plexins [33], integrins [34] and N-cadherin [35], which retain close proximity to GEFs and GAPs. These micro membrane-domains permit the linking of extracellular stimuli to Rho GTPase related signalling pathways (Figure 1). Upon activation, Rho GTPases act on their numerous downstream effectors, including among others serine/threonine kinases, such as Rho-associated protein kinase (ROCK) and protein kinase C-related kinase (PRK) for the Rho subfamily, p21-activated kinase (PAK) and mixed-lineage kinase (MLK) for the Rac subfamily and tyrosine kinases, such as activated Cdc42-associated tyrosine kinase (ACK) for the Cdc42 subfamily. Lipid kinases,—for example, Phosphoinositide 3-kinase (PI3K)—are downstream effectors for Rac and Cdc42 subfamilies and lipases, as well as scaffold proteins, such as diaphanous-related formin-1 (mDia), neutrophil cytosol factor 2 (p67^phox^) and Wiskott–Aldrich Syndrome protein (WASP) for Rho, Rac and Cdc42 subfamilies, respectively (for a review see Reference [36]).

Rho GTPases and in particular RhoA, Rac1 and Cdc42 have been identified and extensively studied as key regulators of actin cytoskeleton assembly and organization [37]. Specifically, RhoA activation promotes the assembly of stress fibres (actin-myosin filaments) and focal adhesions [38], Rac1 induces membrane ruffling and lamellipodia formation [39], whereas Cdc42 activation is responsible for the assembly of filopodia and actin microspikes [40]. Due to their role in actin cytoskeleton dynamics, Rho GTPases have been implicated in many cellular processes that depend on actin cytoskeleton remodelling, such as neuronal axon guidance, phagocytosis, cell migration, cell polarity and cell-cell interactions [41]. Besides microfilaments regulation, Rho GTPases are involved in gene expression and enzymatic activity and, in particular, RhoA, Rac1 and Cdc42 are associated with processes such as apoptosis, cell cycle, reactive oxygen species (ROS) production, membrane trafficking and proliferation [41]. Other processes that Rho GTPases are involved in are cell cycle progression [42], regulation of gene transcription [43] and neuronal morphology [44] as well as neuronal plasticity and migration [45]. As the multifaceted role of Rho GTPases became apparent, dysregulation of their activity was identified in multiple diseases and pathologies, including neurological [46] and neurodegenerative disorders [47,48], inflammation [49] and neuropathic pain [50,51].

In research, inhibitors of the activity of the different member of the Rho GTPases family have been used to understand their implication in all the above mentioned processes (Table 2). Although RhoA, Rac1 and Cdc42 have been identified as important regulators in a plethora of cellular functions in health and disease, the precise molecular pathways and signalling interconnections in the nervous system are complex and in particular in sensory neurons these pathways are not yet fully understood.

## 2. Rho GTPases in Developing and Mature Sensory Neurons

Within the Rho GTPase family, the most studied members in research addressing neuronal development are RhoA, Rac1 and Cdc42 (i.e., References [64,65,66,67,68,69]). They act as intracellular molecular switches that transduce signals from extracellular stimuli to the actin cytoskeleton and the nucleus. These Rho GTPases-associated signals regulate neuronal migration and morphogenesis, including processes, such as axonal polarization, axon growth and guidance, dendrite elaboration and plasticity as well as synapse formation.

Evidence for their importance in sensory neuron development, differentiation and survival is diverse and scattered through all sensory systems, such as cochlear hair cells [70,71], retinal photoreceptors [72] and somatosensory primary afferents including nociceptors [8,73,74]. Novel optogenetic and chemogenetic tools are increasingly available to experimentally target Rho GTPases and are extensively used in retinal and cochlear sensory neurons [75]. In this review we focus on primary somatosensory afferents, since Rho GTPases are recently receiving increasing attention as molecular switches setting the sensitivity to painful stimulation and as promising targets to improve peripheral nerve regeneration.

### 2.1. Expression of Rho GTPases in Sensory Neurons

Research on the role of Rho GTPases in sensory neurons started approximately 25 years ago in model systems as *Caenorhabditis elegans* and *Drosophila melanogaster*. Since the nervous system of *C. elegans* contains only 302 neurons, out of which sensory neurons account for about one-third [76], these animals are perfect models for basic neurobiological analysis. The role of RhoA in the sensory circuitry formation of *C. elegans* during post-embryonic development was described for the first time by W. Chen and L. Lim, who found that RhoA immunoreactivity in sensory neurons as well as in the nerve ring was high during larval development, suggesting a stage-specific role of RhoA in post-embryonic development [73].

In rodents, RhoA expression is particularly high in late embryonic stages and up to postnatal day 1 [74], while modest expression of RhoA mRNA persists during adult age. However, Rho GTPases levels are also profoundly upregulated after injury in sensory neurons found in dorsal root ganglia (DRG) [8], suggesting that Rho GTPases are involved in re- or degenerative processes (see below). Of the Rac type GTPases, Rac1 and Rac3 are expressed during embryonic development in the nervous system including proprioceptive and nociceptive neurons in the DRG [10,77]. Similarly, Cdc42 is expressed in DRG during development and adult age and upregulated following injury [8,77].

### 2.2. Rho GTPases in Sensory Neuron Development

Rho GTPases are essential regulators of the cytoskeleton remodelling, which contributes to several aspects of neuronal development. Once a neuron is born it migrates a long way to find its final destination, where it starts to differentiate. It then sends out two types of processes: several dendrites to collect input and one axon to transport its output to its target cells. After establishing neuronal polarity, the axon navigates through a complex environment to find its destination and also dendrites grow and branch. Finally, synaptic connections to other neurons have to be established [78]. Accumulating evidence suggests that Rho GTPases act as key regulators in several of these processes. Most of the studies have been performed using human cell culture or genetically modified mouse lines. The different steps in neuronal development include specification, branching and elongation as well as retraction, navigation and guidance of axons (see e.g., References [79,80,81,82]), synaptic target side selection, growth and branching of dendrites plus formation and maturation of synapses (see e.g., References [48,83]).

Rho GTPases are involved in the regulation of polarization in different cell types, including neuronal cells. In 2008, Iden and Collard [84] proposed a crosstalk between Rho GTPases and polarity protein members of the partitioning defective (PAR) complex. In more detail, the emergence of a complex consisting of Cdc42, partitioning defective protein 6 (PAR6), PAR3, T cell lymphoma invasion and metastasis 1 (TIAM1) and Rac1 appears to be decisive in establishing neuronal polarity [85,86].

Depletion of RhoA results in severe morphological deficits in the central nervous system (CNS) from embryonic day (E) 11.5, due to the early cell-cycle exit and precocious neuronal differentiation, suggesting an essential role for RhoA in the maintenance of spinal cord neuroepithelium organization and the neural stem cell pool [87]. However, despite the considerably high expression levels, depletion of RhoA from DRG neurons does not lead to major morphological deficits, such as disconnected axons from their target tissues, nor to functional deficits in proprioception or nociception, possibly due to compensatory upregulation of the other family member RhoC [74].

Conditional deletion of Rac1 in the ventricular zone results in neuronal migration deficits; it directs axon guidance, but it is not required for neuritogenesis [88]. In a conditional knock-out mouse, in which Rac1 is ablated in the whole brain, Rac1-deficient cerebellar granule neurons show impaired neuronal migration and axon formation both in vivo and in vitro. In addition, Rac1 ablation disrupts lamellipodia formation in growth cones and abolishes the expression of the WASP family verprolin-homologous protein (WAVE) complex from the plasma membrane of knock-out growth cones [89]. Neuronal death is observed in multiple locations, presumably as a secondary consequence of the axon growth and/or guidance defects. Following deletion of Rac1 in the forebrain, thalamocortical axons were misrouted inferiorly, with the majority projecting to the contralateral thalamus and a minority projecting ipsilaterally to the ventral cortex [90]. A reduction in the number of axons originating from the DRG and the sympathetic chain ganglia, dramatic reduction in the size of the DRG and number of DRG neurons in the brachial and thoracic regions was observed in embryos with a conditional depletion of Rac1, while the expression of DRG markers [brain-specific homeobox/POU domain protein 3A (Brn3a), insulin gene enhancer protein 1/2 (Islet1/2), sex determining region Y-box 10 (Sox10)] appeared unchanged [90]. DRG neurons were affected early in their development, since they were already missing from the sensory ganglia at E11.5 [90]. At present, it is not clear whether the primary defect occurs in neural crest migration, DRG axon growth and/or guidance or a combination of the two. In contrast, no major deficits of peripheral nerve system (PNS) development were observed after depletion of Rac3 [10].

Similar to the reports on RhoA and Rac1, Cdc42 is also involved in neuronal development including proliferation, initial dendritic development and dendritic spine maturation in the CNS [91]. Genetic ablation of Cdc42 in the brain leads to multiple abnormalities, including striking defects in the formation of axonal tracts, which is accompanied by disrupted cytoskeletal organization, enlargement of growth cones and inhibition of filopodial dynamics [92]. In addition, Cdc42 is indispensable for normal DRG development. At later stages than E10.5, Cdc42 conditional knock-out embryos have severe malformations, reminding the phenotype of Rac1 knock-out embryos and DRG are present but underdeveloped and reduced in size [93]. However, DRG cell differentiation seems to not be affected by the Cdc42 loss [93].

### 2.3. Importance of Rho GTPases for Sensory Neuron Survival

In general, Rac GTPases seem to have anti-apoptotic properties promoting neuronal survival acting on two signalling pathways: on one hand by activating the mitogen-activated protein kinase kinase 1/2 (MEK1/2)/extracellular signal–regulated kinase 1/2 (ERK1/2) signalling cascade, which represses the induction of the pro-apoptotic BH3-only protein Bim in an c-Jun N-terminal kinase (JNK)/c-Jun-dependent matter and on the other hand, by inhibiting the Janus kinase (JAK)/signal transducer and activator of transcription 5 (STAT5) signalling cascade that represses anti-apoptotic B-cell lymphoma-extra-large (Bcl-xL) [94,95,96,97]. Conversely, activation of RhoA and/or RhoB and downstream ROCK leads to neuronal apoptosis, which has been documented in different neurodegenerative models [98,99,100]. ROCK inhibitors (e.g. Y-27632) are therefore commonly used in protocols for neuronal differentiation from human induced pluripotent stem cells (iPSCs) to prevent apoptosis and increase survival after stem cell plating and after passaging of early neurons (Table 2) [58,59].

## 3. Rho GTPases in Peripheral Nerve Injury

Peripheral nerves contain sensory, motor and autonomic neurons, as well as non-neuronal cells, such as Schwann cells (SCs), other glial cells (e.g. satellite glial cells) and immune system cells (e.g., macrophages) [101]. Upon a peripheral nerve injury, non-neuronal cells initiate molecular and cellular processes termed Wallerian degeneration at the site of the injury, which together with resident immune cells promote clearing of cell debris and enhance conditions favouring axonal regeneration [102]. RhoA, Rac1 and Cdc42 have been found upregulated and activated in sensory neurons as well as non-neuronal cells following nerve lesion [51,62,103,104]. Therefore, in the following paragraphs we will focus on the role of Rho GTPases in sensory neuron responses to injury and address aspects of neuroregeneration, inflammation and neuropathic pain.

### 3.1. Activation of Rho GTPases in Sensory Neurons After Injury

The different members of the Rho GTPase family can be targeted by soluble factors as well as neighbouring cells to fine tune nerve regeneration after injury. The myelin-associated inhibitors Nogo, myelin-associated glycoprotein (MAG) and oligodendrocyte-myelin glycoprotein (OMgp) are well-known growth inhibitors, which cooperate with members of the Rho GTPases family. MAG is a potent inhibitor of neurite outgrowth localized in SCs. MAG binds both Nogo receptors (NgR), in particular the isoform 2 (NgR2) with high affinity [105]; however, in sensory neurons, deletion of NgRs does not affect the MAG-dependent neurite outgrowth inhibition [106]. Interestingly, the low-affinity neurotrophin receptor p75^NTR^ associates with NgR and acts as a signal transducer for MAG, Nogo and OMgp [107,108]. Activation of p75^NTR^ releases RhoA from RhoGDI and MAG promotes the association of RhoGDI to p75^NTR^, reducing the competitive binding of the RhoGEF Kalirin9 to p75^NTR^ [107]. These interactions cause the activation of RhoA/ROCK and subsequent growth cone collapse and inhibition of sensory axon growth. Moreover, Rho is directly activated by the myelin-associated inhibitor Nogo-66 [56]. Increased levels of active GTP-bound Rho are found in lysates of DRG neurons cultured on Nogo-66 and RhoA/ROCK inhibition promotes neurite outgrowth of sensory neurons in vitro [56].

Bioactive lipids, such as the sphingolipid sphingosine-1-phosphate (S1P) or lysophosphatidic acid (LPA, see below), bind different GPCRs and affect neuroregeneration. S1P_1_ receptor associates with G_i/o_ to activate Rac1 and promotes migration and neurite outgrowth, whereas S1P_2_ and S1P_3_ receptors utilize G_12/13_ to activate RhoA and ROCK, to induce growth cone collapse and inhibit migration [57,109]. An interesting partner, downstream of the RhoA/ROCK signalling, is the collapsin response mediator protein-2 (CRMP2) [110]. CRMP2 plays a role in embryonic development and neuronal polarity and is required for neurite elongation and axon formation [111,112,113]. It binds tubulin heterodimers and promotes microtubules assembly, however, when phosphorylated and inactivated, it inhibits tubulin polymerization and causes cytoskeleton destabilization [114]. ROCK-dependent phosphorylation/inactivation of CRMP2 mediates growth cone collapse and neurite retraction in DRG neurons [57,115,116]. In DRG neuronal cultures, RhoA and ROCK are activated by high doses of S1P through S1P_3_ receptor and mediate phosphorylation of CRMP2 at Thr-555. This post-transcriptional modification inactivates CRMP2 [57] and reduces neurite outgrowth in short-term cultures of both adult DRG neurons and motor neuron-like cells [57]. At very low doses S1P promotes elongation, rather than retraction, probably though S1P_1_ receptor and activation of Rac1 (Figure 2) [57]. In vivo, the lack of S1P_3_ receptors and the consequent RhoA signalling pathway destabilization, promotes functional recovery after peripheral nerve injury [57].

### 3.2. Rho GTPases in Peripheral Neuroregeneration and Neurite Outgrowth

In contrast to neurons in the CNS, axons in the PNS possess the potential to regenerate. Rho GTPases regulate processes that are essential for the survival of neurons after injury and the subsequent target reinnervation. Dendritic arborization, spine formation, growth cone development and axon guidance are critically mediated by Rho GTPases. A tight spatial and temporal regulation of the Rho GTPases family members is crucial for proper neuronal morphology and nerve fibre regeneration after injury. This regulation is attributed to the spatio-temporal activation of GEFs and GAPs [117]. Additionally, antagonistic effects between the different members of the Rho GTPase family affect axon growth, axonal branching and growth cone formation, which are some of the most important processes of regeneration after injury [118]. Rac1/Cdc42 promote neuroregeneration, whereas Rho negatively affects actin dynamics, cellular shape and motility [41]. Despite the increased PNS plasticity and neuronal regeneration capacity, compared to CNS, functional reinnervation in adulthood by injured peripheral neurons is often difficult and not perfectly completed [41,88,119,120,121]. Immediately after injury, regenerating and uninjured axons compete to grow into the denervated tissue, such as the skin. In more advanced stages of regeneration, the capacity of the uninjured neurons to sprout and occupy the injured territories drastically diminishes. Studies on regeneration of trigeminal sensory axon terminals in live zebrafish larvae after axotomy showed that even regenerating injured axons might be repelled by the denervated skin at later stages of target reinnervation [121]. Interestingly, those regenerating axons are repelled by their former territories, where local inhibitory factors, like members of the NgR/RhoA pathway, persist in their inhibitory function. Indeed, NgR, its co-receptor leucine rich repeat and immunoglobin-like domain-containing protein 1 (LINGO-1), RhoA, ROCK and their intracellular partner CRMP2 might be partially required for the avoidance of these skin former territories and O’Brien and colleagues demonstrated that antagonizing these factors improves the ability of injured axon to successfully reinnervate the skin after axotomy [121].

While RhoA elicits the assembly of actin stress fibres and focal adhesion and causes neurite retraction, Rac1 controls lamellipodia formation and membrane ruffles, through actin filament accumulation at the cell membrane, whereas Cdc42 stimulates filopodia formation and neurite outgrowth [117]. Dominant negative Rac1 in embryonic sensory neurons leads to axonal outgrowth defects without affecting dendrite growth, whereas Cdc42 mutations affect both axons and dendrites [122]. Mechanical tension appears closely linked to the development of neurites and defective neurite formation is observed in Rac1 deficient cells. The reason for this phenomenon seems to be a combination of compromised adhesion and motility in response to the lack of Rac1 (e.g., Reference [34]) and the organization of the actin cytoskeleton. Similar observations were described by Kozma et al. [44]. This and other studies in neuronal-like cells suggest that the fine balance between the outgrowth of neurites, the retraction of neurites and the increased activity of filopodium and/or lamellipodium at the growth cone might be regulated by the interplay between Rac1, Cdc42 and RhoA [44,123]. On the other hand, inhibition of RhoA by the recombinant membrane permeable *Clostridium botulinum* C3 toxin (BoTXC3) produces a minor but sufficient, outgrowth effect on the axon of peripheral sensory neurons after injury [55]. Small peptides obtained from BoTXC3 promote axon regeneration and motor recovery in both injured CNS and PNS [124,125]. However, direct neuronal overexpression of C3 transferase does not enhance axonal growth [55].

Growth and regeneration of sensory axons are also controlled via PI3K pathway, in which Akt regulates actin cytoskeleton via Rac1 and microtubules dynamics through the inactivation of glycogen synthase kinase 3 beta (GSK-3β) [126,127]. Indeed, PI3K regulates various downstream partners that are important for axon polarity establishment and maintenance. PI3K is the so called symmetry-breaking signalling molecule and its activation guides the development of one of the neurites into an axon. Its overexpression enhances axon calibre and branching, likewise its inhibition was shown to inhibit axon formation [128,129,130]. Upon NGF stimulation, PI3K increases in fact Rac1 activity and transiently decreases RhoA signalling in the first stage of neurite outgrowth [55,131]. The PI3K-linked kinase p110δ is widely expressed in the PNS and its inactivation leads to an increased vulnerability of DRG neurons to growth cone collapse and decreased axon elongation [132]. In adult mice, loss of p110δ reduced axon regeneration and functional recovery after sciatic nerve injury and this impairment was associated with Akt signalling reduction and, noteworthy, RhoA activation [132]. Restoration of axonal extension in DRG neurons was achieved through pharmacological inhibition of the downstream signalling partner ROCK [56,62,132].

Moreover, the ROCK inhibitor fasudil increases axon numbers and improves axonal regeneration after sciatic nerve injury in vivo [62,63]. The response of sensory and motor nerves to peripheral nerve crush varies depending by the activation levels of RhoA and ROCK. In the presence of a non-permissive growth environment, sensory neurons cultured on chondroitin sulphate proteoglycans (CSPGs) were less responsive to the ROCK inhibitor Y-27632 compared to motor neurons [133]. These differences in response were associated to altered RhoA activation. In vivo, ROCK inhibition enhanced the regeneration of motor axons, whereas the growth of sensory fibres was not really affected [133]. Thus, the fine tuning of the RhoA/ROCK pathway may differently influence the capacity of sensory and motor neurons to regenerate. These discrepancies might be attributed either to the organism developmental stage or the cell type-specific regulation of Rho GTPase expression or, as mentioned above, the activation of specific receptors.

### 3.3. Rho GTPases in Neuroinflammation

Most of the current literature highlights the role of RhoA on axonal regeneration and neuronal survival, however, there are several studies underlining the importance of Rho GTPases in SCs during regeneration. The Rho GTPases, in particular RhoA, regulate SCs differentiation, proliferation, migration and myelination mediated by the downstream JNK pathway and the p38 (MAPK) [134]. Interestingly, high levels of RhoA and Rac1 mRNA are found in SC derived exosomes, suggesting a role of these GTPases as cargo for intercellular signalling [135]. Cdc42 mRNA and protein levels increase significantly after sciatic nerve injury and a drastic reduction in Cdc42 mRNA expression inhibits SCs proliferation and migration [12,136]. Thus, SCs appear to support neuronal regeneration and proper target reinnervation through Rho GTPases. Furthermore, upon peripheral nerve injury, SCs, as well as resident and migrating immune cells, release pro-inflammatory mediators such as prostaglandins, reactive oxygen species (ROS), nitric oxide (NO), interleukins (IL) -1β (IL-1β), IL-6, IL-8, IL-18, tumour necrosis factor α (TNF-α) and leukaemia inhibitory factor (LIF), as well as anti-inflammatory cytokines, such as IL-10 and transforming growth factor β1 (TGF-β1) [137,138,139]. RhoA [140,141], Rac1 [142,143] and Cdc42 [141] activate in response to cytokines and lipopolysaccharide (LPS). Additionally, ROCK1 overexpression and LPS treatment promote inflammation and apoptosis via toll-like receptor 4 (TLR4) signalling, MAPKs (p38, ERK and JNK) activation, nuclear factor kappa-light-chain-enhancer of activated B cells (NF-κB) activation and increased expression of IL-6, IL-8, IL-1β and TNF-α [144,145]. The RhoA/ROCK pathway is commonly found upregulated in inflammatory processes and the inhibition of ROCK is usually associated with anti-inflammatory effects. Interestingly, in a rat model for inflammation, local application of low doses of the ROCK inhibitors Y-27632 and fasudil mediated pro-nociceptive responses, whereas high doses induced hypoalgesia and reduced paw oedemas [60]. Furthermore, the TNF-α-Rac1-NF-κB axis activation induced the expression of matrix metalloproteinase 9 (MMP9) and inhibition of this signalling pathway resulted in anti-inflammatory effects [146]. In macrophages during nerve re-myelination in the area of the peripheral nerve injury, RhoA is activated via repulsive interactions between myelin, MAG, present on SCs and macrophage Nogo receptors 1 and 2 [104]. The activation of RhoA/ROCK reduces macrophage adhesion and enhances clearance from the site of inflammation through SC basal lamina, contributing to the termination of the inflammatory response [104].

### 3.4. Rho GTPases and Nociception

Due to their role in actin organization RhoA/ROCK affect ion channel and receptor membrane localization. Injection of complete Freunds’ adjuvant into the paw induces sensitization of nociceptors to mechanical stimuli and mechanical hypersensitivity by modulating the function and membrane availability of transient receptor potential cation channel, subfamily A, member 1 (TRPA1). This process occurs through semaphorin 4C (Sema4C)/Plexin-B2/RhoA/ROCK signalling and the hypersensitivity was abolished by treatment with the ROCK inhibitor Y-27632 [33]. In the same model, heat-shock cognate 70 (Hsc70), a member of the heat shock protein (Hsp) chaperones, was found to promote the removal of the transient receptor potential vanilloid type 1 (TRPV1) from the cell membrane of DRG neurons by inhibiting the ROCK-dependent phosphorylation of TRPV1 [147].

Neuropathic pain is a consequence of nerve injury [148] and Rho GTPAses and their effectors have been implicated in the development and maintenance of neuropathic pain [50,51,149,150,151,152,153,154]. In this context the RhoA/ROCK pathway has been the most intensely studied and associated with aggravated behavioural and physiological responses in animal models of peripheral neuropathic pain, such as the partial sciatic nerve injury [50,149,151], the chronic constriction injury (CCI) [39], the spared nerve injury (SNI) [152], the spinal nerve transection [153] and the spinal nerve ligation (SNL) [150,154]. The Rac1 subfamily of Rho GTPases seems to also be involved in neuropathic pain processes [150,155].

In addition to the aforementioned S1P, another bioactive lipid LPA is released by activated platelets after tissue injury and signals through GPCR receptors LPA_1_, LPA_2_, LPA_3_ and LPA_4_ [156]. Of particular interest in neuropathic pain research is the LPA_1_ receptor, which upon binding LPA or lysophosphatidylcholine (LPC) signals via Gα_12/13_ proteins and induces RhoA activation [50,157]. Additionally, LPA potentially via the RhoA/ROCK pathway promotes the demyelination of A-fibres in vivo and in ex vivo dorsal root fibre cultures [50,158]. The LPA-induced, as well as injury-induced allodynia, hyperalgesia and demyelination and voltage-gated calcium channel α2δ1 subunit (Ca_α2δ1_) upregulation in the DRG, were abolished by either depleting the LPA_1_ receptor or by inhibition of RhoA or ROCK [50]. LPA- or nerve injury-induced pain is reduced by preventive inhibition of LPA_1_-RhoA-ROCK, suggesting that this signalling pathway is more important in the initiation but not maintenance of neuropathic pain [50]. The LPA-RhoA pathway has been found to upregulate 82 genes in DRG, including Ephrin B1 and calcium/calmodulin-dependent protein kinase II alpha (CAMKIIα), which may also contribute to this process [159]. In particular, ephrin receptors appear to play a role in neuropathic pain, since deletion of EphB2 in voltage-gated sodium ion channel 1.8-positive (Nav1.8+) nociceptive sensory neurons disrupted the increased thermal hyperalgesia and mechanical allodynia in mice subjected to a partial sciatic nerve injury [160].

Inhibition of Rho GTPase isoprenylation promotes anti-inflammatory and anti-nociceptive effects. Statins, such as simvastatin, are 3-Hydroxy-3-methylglutaryl coenzyme A (HMG-CoA) reductase inhibitors, which are commonly used for lowering cholesterol but also exert neuroprotective pleiotropic effects, including anti-inflammatory and anti-nociceptive actions [53]. Studies suggest that statins exhibit their effects by preventing the isoprenylation of Rho GTPases and thus inactivating them [53,54]. Simvastatin exhibited its effects on RhoA by retaining it inactive in the cytoplasm and this alleviated thermal hyperalgesia induced by experimental nerve injury [53,161]. Increased mechanical and thermal hypersensitivity are associated with increased RhoA membrane translocation, ROCK activation and subsequent LIM domain kinase (LIMK) and cofilin phosphorylation in isolectin B4 (IB4) and calcitonin gene-related peptide (CGRP) positive DRG neurons. Simvastatin attenuated pain behaviours and reduced RhoA membrane localization, thus inhibiting the ROCK/LIMK/cofilin pathway [51]. Similar effects are reported upon inhibition of either ROCK or cofilin phosphorylation [51]. Furthermore, the RhoA/ROCK/LIMK/cofilin signalling pathway has been implicated in the trafficking of delta opioid receptor (δOR) from the Golgi apparatus to the cell membrane of DRG neurons through a beta-arrestin 1 (β-arr1)-dependent mechanism and it has been proposed that this pathway could be involved in the trafficking of other GPCRs, such as dopamine receptor 1 (DA1), neurotensin or protease-activated receptor 2 (PAR2) [162]. Although the DRG and primary afferent nociceptors are considered to be the initial station for the processing of painful stimuli, Rho GTPases and the underlying molecular interconnections are most relevant at the spinal dorsal horn level, which is not addressed in this review.

## 4. Conclusions

In sensory neurons and conditions related to neuronal development, differentiation, migration and regeneration, RhoA limits these processes and reduces regeneration through growth cone collapse and neurite retraction, while Rac1/Cdc42 promote a favourable environment for neuronal regeneration. Rho GTPases appear to be critically important for many processes following peripheral sensory neuron lesions and the signalling pathways they initiate and crosstalk with, are directly involved in inflammatory responses and neuropathic pain, indicating for example, that RhoA and its downstream effectors could serve as targets for therapeutic interventions. Although this review addressed all available information on the role of Rho GTPases in peripheral sensory neurons, the complexity and bidirectional interaction between inflammation and neuropathic pain in combination with the multidimensional roles of Rho GTPases and the lack of knowledge on the functions and interactions of most of the other members of the family in peripheral sensory neurons, necessitate further research in order to elucidate the molecular and physiological aspects, as well as the therapeutic potential, of these small intriguing proteins. Since activation of Rho GTPases occurs through numerous different mediators, revealing the nature of the complex signal through which Rho, Rac and Cdc42 fine tune growth cone extension and retraction may be difficult. Although the above studies and considerations indicate that Rho GTPases are remarkable targets of clinical importance, targeting upstream processes setting Rho GTPase activation tailored to the specific condition appears to be a more favourable strategy in order to add specificity and balance the activity of Rho GTPase signals under specific conditions.

## Figures and Tables

**Figure 1 cells-08-00591-f001:**
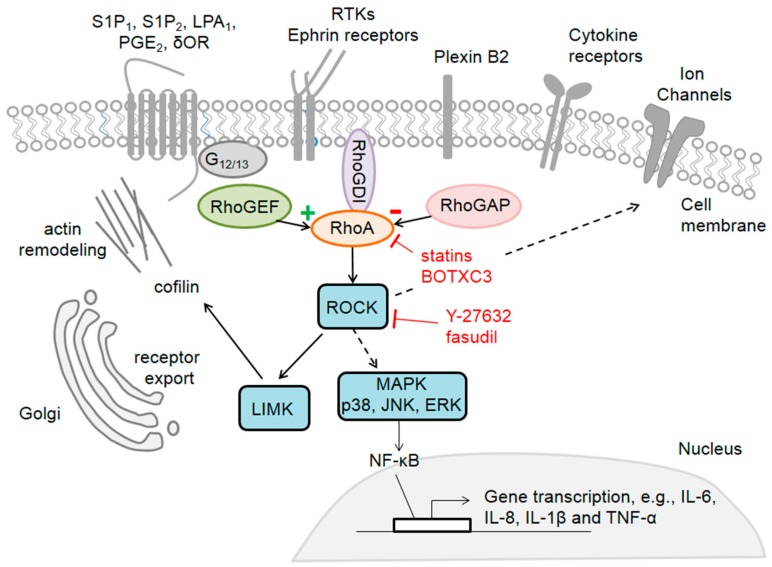
The family member RhoA, effectors and activators in sensory neurons. RhoA can be activated via different receptors, such as GPCRs (G-protein coupled receptors), RTKs (receptors of the tyrosine kinase family), cytokine receptors or ion channels. RhoA activity is regulated by specific proteins: GEFs (guanine nucleotide exchange factors) which promote its active state and GAPs (GTPase activating proteins) which turn it into an inactive state. By regulating various downstream effectors, RhoA elicits changes in the actin cytoskeleton and these effects can be pharmacologically modulated by inhibitors like *Clostridium botulinum* C3 toxin (BoTXC3) or fasudil.

**Figure 2 cells-08-00591-f002:**
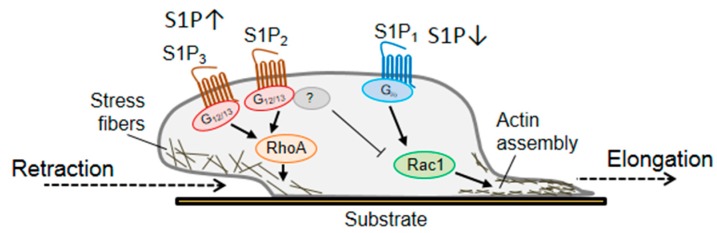
The sphingolipid S1P and the involvement of Rho GTPases in the outgrowth of sensory neurons. The bioactive lipid S1P elicits its action in sensory neurons depending on its local concentration and activates different members of the Rho GTPases family. High levels of S1P activate RhoA through the S1P_3_ receptor and sensory neurons respond with a rapid retraction of neurites and growth cone collapse. On the other hand, S1P_1_ receptor seems to be associated with Rac1 activation and consequent elongation of neuronal processes.

**Table 1 cells-08-00591-t001:** The family of Rho GTPases, their members and expression in peripheral sensory neurons.

**Typical Rho GTPases**					
**Subfamily**	**Members**	**Expressed in peripheral sensory neurons**	**Studied in PNI**	**Expressed in other neuronal cells**	**Reference**
Rho	RhoA	Yes	Yes	Yes	[8]
	RhoB	Yes	No	Yes	[8]
	RhoC	Yes	No	Yes	[8]
Rac	Rac1	Yes	Yes	Yes	[8]
	Rac2	Yes	No	Yes	[9]
	Rac3	Yes	No	Yes	[10]
	RhoG	not documented	No	Yes	[11]
Cdc42	Cdc42	Yes	Yes	Yes	[8,12]
	RhoQ (TC10)	Yes	No	Yes	[8]
	RhoJ (TCL)	not documented	No	No	
RhoF/RhoD	RhoF (Rif)	not documented	No	Yes	[13]
	RhoD	not documented	No	Yes	[14]
**Atypical Rho GTPases**					
**Subfamily**	**Members**	**Expressed in peripheral sensory neurons**	**Studied in PNI**	**Expressed in other neuronal cells**	**Reference**
Rnd	Rnd1 (RhoS)	not documented	No	Yes	[15]
	Rnd2 (RhoN)	not documented	No	Yes	[16]
	Rnd3 (RhoE)	not documented	No	Yes	[17]
RhoBTB	RhoBTB1	not documented	No	Yes	[18]
	RHoBTB2				
RhoH	RhoH (TTF)	not documented	No	No	
RhoU/RhoV	RhoU (Wrch1)	not documented	No	Yes	[19]
	RhoV (Chp/Wrch2)				

PNI: peripheral nerve injury.

**Table 2 cells-08-00591-t002:** The most commonly used inhibitors of Rho GTPases and downstream partners used in studies on peripheral sensory neurons.

Inhibitor	Target	Role in Physiology	Role in Pathophysiology
statins e.g., simvastatin	prevents the isoprenylation of Rho GTPases	Promoted neurogenesis and migration of neural stem cells [52]	Attenuated pain behaviours after PNI [51,53,54]
BoTXC3	RhoA RhoB RhoC		Attenuated pain behaviours after PNI [50] Promoted outgrowth of DRG [55,56,57]
Y-27632	ROCK	Promoted neuronal differentiation of iPSCs [58,59] Promoted neurogenesis and migration of neural stem cells [52]	Promoted DRG outgrowth in vitro [56] Attenuated pain behaviours after PNI [33,50] Low doses mediated pro-nociceptive responses [60] High doses induced hypoalgesia and reduced paw oedema [60]
fasudil (HA-1077)	ROCK	Increased neurite outgrowth of GFRα1 DRG neurons in vitro [61]	Promoted DRG outgrowth in vitro [62] Improved axonal regeneration after PNI [62,63] Low dose mediated pro-nociceptive responses [60] High doses induced hypoalgesia and reduced paw oedema [60]

BoTXC3: *Clostridium botulinum* C3 toxin; DRG: dorsal root ganglia; GFR α1: glial cell–derived neurotrophic factor coreceptor α1; iPSCs: induced pluripotent stem cells; PNI: peripheral nerve injury.

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
