# Peer review of "Rho GTPases in the Physiology and Pathophysiology of Peripheral Sensory Neurons"

_cells, 2019, doi:10.3390/cells8060591_

Round 1
Reviewer 1 Report
This review from Kalpachidou et al. is precise and up to date. The overall construction is well organized, and several aspects of the field are treated with seriousness.
I have only few concerns:
1- Line 64: the authors describe PRK, an effector of Rho as “protein kinase R”. This is false, in fact, it is the “PKC-related protein kinase”. Please modify.
2- Line 85After describing the effects on actin cytoskeleton, the authors begin the next sentence by: “Beside microtubules…”. I think it’s a mistake (it should be “microfilaments” instead, but if not (and maybe they wanted to talk about Rho GTPases effects on microtubule, then it should be developed.
3- Line 122: to make understand that you describe a general level of all RhoGTPases and not only RhoA, please add GTPases after Rho in the end of the line (before levels)
4- Line 129: again I think this is a mistake: cytokinesis is misused here. You certainly aimed to talk about migration or chimiotactism or even growth cone progression. Please modify.
5- Lines 281-287: This point is very interesting and could be developed. Several studies describe the role of PI3K on polarity establishment and maintenance, in relation with RhoGTPases. You could add a sentence on such function.
The conclusion is very wise and one can hope it will avoid misconceived clinical studies.
Author Response
We thank the reviewer for his/her very valuable suggestions.
Reviewer #1
This review from Kalpachidou et al. is precise and up to date. The overall construction is well organized, and several aspects of the field are treated with seriousness.
I have only few concerns:
1-Line 64: the authors describe PRK, an effector of Rho as “protein kinase R”. This is false, in fact, it is the “PKC-related protein kinase”. Please modify.
We apologize for the profound mistake and thank the reviewer for pointing it out. The text has been corrected in the revised manuscript.
2-Line 85: After describing the effects on actin cytoskeleton, the authors begin the next sentence by: “Beside microtubules…”. I think it’s a mistake (it should be “microfilaments” instead, but if not (and maybe they wanted to talk about Rho GTPases effects on microtubule, then it should be developed.
We agree with the reviewer and the text has been modified in the revised manuscript with the correct wording “microfilaments”.
3-Line 122: to make understand that you describe a general level of all RhoGTPases and not only RhoA, please add GTPases after Rho in the end of the line (before levels)
We thank the reviewer for the insightful comment. The text has been corrected in the revised manuscript.
4-Line 129: again I think this is a mistake: cytokinesis is misused here. You certainly aimed to talk about migration or chimiotactism or even growth cone progression. Please modify.
We apologize for the profound mistake and thank the reviewer for pointing it out. The text has been corrected in the revised manuscript.
5-Lines 281-287: This point is very interesting and could be developed. Several studies describe the role of PI3K on polarity establishment and maintenance, in relation with RhoGTPases. You could add a sentence on such function.
We agree with the reviewer that this point is very important to be developed, indeed PI3K is involved in the breaking of the cellular symmetry, a very central process in neuronal polarity. We added the required information at lines 296-301 (in the tracked mode version).
The conclusion is very wise and one can hope it will avoid misconceived clinical studies.
We thank the reviewer for this comment and we appreciate it.

Reviewer 2 Report
The review article by Kalpachidou et. al entitled “RhoGTPases in the physiology and pathophysiology of peripheral sensory neurons” is an interesting literature review that is worth publishing.
It is well-written, includes sub-sections that make reading easier and all the necessary information on the topic.
There are a few minor things that need to be corrected though:
1) The authors should define all abbreviations the first time they appear in the text and then refer to them using the abbreviated name i.e. Cdc42, ROCK, PI3K etc. This should be applied throughout the text including the abstract.
2) In line 32, it is written that RhoGTPases hydrolyze GDP to GTP while the opposite is true
3) In table 1 the authors should consider including two more columns indicating by a tick, for instance, whether the specific Rho GTPase is expressed in other neuronal cells or whether it has been studied in peripheral nerve injury (instead of using * and #).
4) Also, it might have been nice if there was another table summarizing the different inhibitors of RhoGTPases and their effect in peripheral sensory neurons’ physiology/pathophysiology.
5) The authors should consider rephrasing lines 396-397 to convey the notion that in this review you covered all there is on the topic but there are still things/questions that need to be addressed.
6) The last sentence of the abstract needs to be revisited as there are grammar-syntax errors.
Author Response
We thank the reviewer for his/her very valuable suggestions.
Reviewer #2
The review article by Kalpachidou et. al entitled “RhoGTPases in the physiology and pathophysiology of peripheral sensory neurons” is an interesting literature review that is worth publishing. It is well-written, includes sub-sections that make reading easier and all the necessary information on the topic.
There are a few minor things that need to be corrected though:
1) The authors should define all abbreviations the first time they appear in the text and then refer to them using the abbreviated name i.e. Cdc42, ROCK, PI3K etc. This should be applied throughout the text including the abstract.
We thank the reviewer for meticulously reading our manuscript. We have defined in the revised manuscript all abbreviations upon their first appearance.
2) In line 32, it is written that RhoGTPases hydrolyze GDP to GTP while the opposite is true
We apologize for the profound mistake and thank the reviewer for pointing it out. The text has been corrected in the revised manuscript.
3) In table 1 the authors should consider including two more columns indicating by a tick, for instance, whether the specific Rho GTPase is expressed in other neuronal cells or whether it has been studied in peripheral nerve injury (instead of using * and #).
We thank the reviewer for the suggestion. We have included the two additional columns in the revised manuscript.
4) Also, it might have been nice if there was another table summarizing the different inhibitors of RhoGTPases and their effect in peripheral sensory neurons’ physiology/pathophysiology.
We thank the reviewer for the suggestion. We have included a table (Table 2) and the corresponding information in the revised manuscript. We agree with the reviewer that an overview on the different inhibitors and their effect could be of significant interest for the reader.
5) The authors should consider rephrasing lines 396-397 to convey the notion that in this review you covered all there is on the topic but there are still things/questions that need to be addressed.
We thank the reviewer for the insightful suggestion. We have rephrased the text in the revised manuscript.
6) The last sentence of the abstract needs to be revisited as there are grammar-syntax errors
We thank the reviewer for the comment. We have revised the sentence accordingly.
